# QuantiFERON CMV Test and CMV Serostatus in Lung Transplant: Stratification Risk for Infection, Chronic and Acute Allograft Rejection

**DOI:** 10.3390/v16081251

**Published:** 2024-08-04

**Authors:** Paolo Solidoro, Federico Sciarrone, Francesca Sidoti, Filippo Patrucco, Elisa Zanotto, Massimo Boffini, Rocco Francesco Rinaldo, Alessandro Bondi, Carlo Albera, Antonio Curtoni, Cristina Costa

**Affiliations:** 1Division of Respiratory Medicine, Cardiovascular and Thoracic Department, AOU Città della Salute e della Scienza di Torino, 10126 Torino, Italy; psolidoro@cittadellasalute.to.it (P.S.); fedesciarrone@gmail.com (F.S.); roccofrancesco.rinaldo@unito.it (R.F.R.); carlo.albera@unito.it (C.A.); 2Medical Sciences Department, University of Turin, 10126 Torino, Italy; 3Division of Virology, Department of Public Health and Pediatrics, AOU Città della Salute e della Scienza di Torino, 10126 Torino, Italy; francesca.sidoti@unito.it (F.S.); elisa.zanotto@unito.it (E.Z.); 4Respiratory Diseases Unit, Medical Department, AOU Maggiore della Carità di Novara, 28100 Novara, Italy; 5Cardiac Surgery Division, Surgical Sciences Department, AOU Città della Salute e della Scienza di Torino, University of Turin, 10126 Torino, Italy; massimo.boffini@unito.it; 6Division of Virology, Department of Public Health and Pediatrics, AOU Città della Salute e della Scienza di Torino, University of Turin, 10126 Torino, Italy; alessandro.bondi@unito.it (A.B.); antonio.curtoni@gmail.com (A.C.); cristina.costa@unito.it (C.C.)

**Keywords:** lung transplantation, CMV, cytomegalovirus, QuantiFERON, rejection

## Abstract

The QuantiFERON CMV (QCMV) test evaluates specific adaptive immune system activity against CMV by measuring IFN-γ released by activated CD8+ T lymphocytes. We aimed to evaluate the QCMV test as a predictive tool for CMV manifestations and acute or chronic lung allograft rejection (AR and CLAD) in lung transplant (LTx) patients. A total of 73 patients were divided into four groups based on donor and recipient (D/R) serology for CMV and QCMV assay: group A low-risk for CMV infection and disease (D−/R−); group B and C at intermediate-risk (R+), group B with non-reactive QCMV and group C with reactive QCMV; group D at high-risk (D+/R−). Group D patients experienced higher viral replication; no differences were observed among R+ patients of groups B and C. D+/R− patients had a higher number of AR events and group C presented a lower incidence of AR. Prevalence of CLAD at 24 months was higher in group B with a higher risk of CLAD development (OR 6.33). The QCMV test allows us to identify R+ non-reactive QCMV population as the most exposed to onset of CLAD. This population had a higher, although non-significant, susceptibility to AR compared to the R+ population with reactive QCMV.

## 1. Introduction

Cytomegalovirus (CMV) replication and infectious disease play a central role in lung transplantation (LTx) complications. In terms of LT outcome, the prevention of short-term and late complications such as acute rejection (AR) or bronchiolitis obliterans syndrome (BOS) due to CMV replication is of the utmost importance [1,2,3,4].

Pre-transplant immunological CMV serostatus is defined by evaluation of CMV gamma globulines (IgGs) of the donor (D) and recipient (R); the risk of CMV infectious events in LTx is based on CMV D and R serostatus: low risk with both negative CMV IgGs in D and R (D−R−), intermediate risk in case of R with specific IgGs (D+R+ or D−R+), and high risk with CMV IgGs D+ and R−.

Moreover, CMV serostatus can predict not only the length of the prophylaxis therapy but also a more careful virological monitoring to prevent CMV infection or disease, as suggested in recent guidelines [5].

Evaluation of CMV DNA in peripheral blood or broncholaveolar lavage (BAL) specimens by molecular assays is usually performed to monitor CMV replication [5]. QuantiFERON CMV assay (QCMV) is an emerging tool used to identify patients with a higher risk of CMV replication. This assay measures interferon-γ (IFN-γ) released by T lymphocytes reacting against CMV; therefore, this test predicts the activity of cellular immune defense against CMV.

Some authors found that a non-reactive QCMV is strictly related to a higher incidence of CMV infection and/or disease [5,6]. Moreover, the QCMV test helps us to identify patients at risk of CMV replication and who are therefore susceptible to a more prolonged prophylactic therapy [7,8]. Within patients at moderate-severe risk (R+, D+R−), a non-reactive QCMV is associated with an increase in viral replication compared to the population with a reactive QCMV [9]. Previous immune response derived from a CMV infection or a consequent cellular immunity activation against CMV represent a protective factor regarding viral replication during follow-up [10].

Herein, we aimed to evaluate a possible predictive and/or prognostic role for QCMV assay in R+ patients universally considered to be at intermediate risk. The primary end point of the study was to evaluate the effect of CD8+ CMV-specific cellular immunity identified by QCMV response on chronic lung allograft rejection (CLAD)-related mortality at two years after transplantation. Secondary outcomes consisted of the evaluation of the role of CMV-specific cellular immunity in acute rejection and in viral replication, both in the blood and in the alveolar microenvironment.

## 2. Materials and Methods

We conducted a single-center retrospective study analyzing clinical microbiological parameters and rejection-related data of patients undergoing LTx at the Lung Transplant Center of the University Hospital Città della Salute e della Scienza di Torino, Turin, Italy, over the period between July 2016 and February 2023. This study was approved by the institutional review board (Protocol No. 0004577—CS/416).

### 2.1. Data Collection

All the data were gathered and collected from the ITR02 database used for scheduled pre- and post-transplant evaluations (ITR02, Piedmont and Aosta Valley Regional Transplant Information System, Oracle Fusion Middleware).

For each patient, the following data were collected: demographic features, date of transplantation, donor and recipient CMV serostatus, data referable to AR and CLAD development, mortality after 24 months post-transplantation (from any LTx related cause), an CMV viral replication on blood and bronchoalveolar lavage (BAL) specimens during the follow-up.

After transplantation, each patient was regularly evaluated at months 1, 4, 8, 12, 18, and 24; the follow-up protocol included evaluation of CMV DNA by PCR on whole blood and BAL specimens and examination of transbronchial biopsies (TBB) for identification of AR. Preliminarily, a complete spirometry with alveolar-capillary diffusion and a high-resolution chest CT were scheduled for each patient.

### 2.2. CMV Prophylaxis and Therapy

According to our center routine practice, a combined universal CMV prophylaxis scheme [11], including immunoglobulin IgG infusion with anti-CMV intravenous antibodies on postoperative days 1, 4, 8, 16, and 30 with a dosage of 0.75 mg/kg of body weight is performed in all patients (500 U of CMVIG, Cytotect Biotest, are composed of IgG1 62%, IgG2 34%, IgG3 0.5%, IgG4 3.5%, and immunoglobulin A (IgA) 5 mg). Subsequently, these infusions are performed monthly until the 18th month post-transplant at a dosage of 0.5 mg/kg. The remaining antiviral therapy involves the use of ganciclovir ev at a dosage of 5 mg/kg twice a day (adjusted according to renal function) from the 15th post-transplantation day for a duration of 3 to 4 weeks, depending on CMV DNA results on whole blood and/or BAL. In case of home discharge or good clinical condition, a shift to valganciclovir per os is performed. Cut-offs for initiation of therapy with intravenous ganciclovir or oral valganciclovir during the follow-up period are as follows: CMV DNA load greater than 10^5^ copies/mL and/or greater than 10^4^ copies/mL on whole blood and BAL, respectively [12,13]. CMV genome load on BAL specimen standardization was conducted according to previous work and was detailed elsewhere [13]. For the treatment of CMV infection, intravenous ganciclovir at 5 mg/kg of body weight twice a day or oral valganciclovir 450 mg twice a day are used; in the occurrence of CMV disease, the dosage of valganciclovir is doubled unless adjusted according to renal function.

### 2.3. Sample Analysis

The quantitative evaluation of CMV DNA on whole blood and BAL, as well as specific anti-CMV therapy and ARs, were studied by collecting cumulative data up to months 12, 18, and 24 of follow-up. The QCMV assay can result as “reactive” (≥0.20 IU/mL), “non-reactive” (<0.20 IU/mL), or “indeterminate” on the basis of IFN-γ production at both sample-test and sample-control stimulated with mitogenic agent (outcome usually related to test preparation errors or poor lymphocyte population on blood drawn) [14].

CLAD was diagnosed after evidence of functional decline greater than or equal to 20% compared to best FEV1, according to the criteria defined by Cooper et al. [15].

CLAD events and all causes mortality were compared at 24 months post-transplantation among different groups.

### 2.4. Study Population

The QCMV assay was performed for all patients between months 4 and 8 following transplantation; non-reactive QCMV results were repeated at the next follow-up time. Patients with just one non-reactive QCMV result were excluded from the study. Each patient was followed for at least 12 months after transplantation; patients who died before 12 months of follow-up or patients alive with a follow-up of less than 12 months were excluded from the study.

On the basis of donor and recipient serology for CMV in the pre-transplantation period, and on the basis of the QCMV assay results, patients were divided into four groups: group A at low-risk (D−R−) with at least two follow-up detections of QCMV persistently non-reactive (D−R−Q−); group B at intermediate risk (R+) with at least two consecutive detections of QCMV persistently non-reactive (R+Q−); group C at intermediate risk (R+) with reactive QCMV from the first measurement or following conversion (R+Q+); group D at high risk (D+R−) with non-reactive or reactive QCMV.

### 2.5. Statistical Analysis

Data were presented as proportions and percentages for categorical variables, as well as means for continuous variables.

For statistical analysis, chi-square test and Fisher’s test were used for differences between categorical variables, and two-tailed *t*-student test and Mann–Whitney test for parametric and nonparametric continuous variables, respectively. Analysis of variance (ANOVA test) for single variable was used for comparison of parametric sample means between more than 2 groups. Survival analysis with Kaplan–Meier plot was evaluated by survival distribution between two groups (group B and C) with log-rank test. All analyses were performed using the Prism 7.0 software (GraphPad, La Jolla, CA, USA). A probability (*p*) value less than 0.05 is considered for the statistical difference.

## 3. Results

Overall, 73 patients belonging to different groups were included in the study: 8.2% group A (six patients), 12.3% group B (nine patients), 65.8% group C (48 patients), and 13.7% group D (10 patients). Demographic data are presented in Table 1 (Table 1).

The average value of QCMV was different in all groups. Although groups A and B included both non-reactive QCMV patients, there was also a difference in the mean test value between groups A and B. There was no difference between the groups regarding the timing of the first test dosage.

The evaluation of end-stage lung diseases indication for LTx was statistically non-significant between groups, except for a higher percentage of the population affected by cystic fibrosis in groups A and D (*p* < 0.05).

Virological data of CMV DNA on whole blood and BAL, initiation of specific anti-CMV therapy in the presence of viral replication or when clinically indicated, and the occurrence of CMV pneumonitis are reported in Table 2 (Table 2).

Overall, the occurrence of CMV replication on whole blood evaluated at different levels of viral load (i.e., >10^5^ copies/mL, 10^4^ and 10^3^) evidenced no differences between groups. Group D showed a higher viral load on BAL than other groups, even if in Group A, no occurrence of CMV replication was evidenced. In accordance with the occurrence of CMV replication on BAL, group D had a higher rate of CMV treatments. No case of CMV pneumonitis was diagnosed among the groups.

As expected, no episode of CMV on blood or BAL was found in group A (D−/R−Q−).

Data regarding the amount of AR at 12, 18, and 24 months were investigated. CLAD (with relative difference in days from transplantation to diagnosis) and death at 24 months after transplantation were also reported (Table 3).

D+R− patients (Group D) showed a higher incidence of ARs (month 12: *p* < 0.05; month 18: *p* < 0.01; month 24: *p* < 0.01). In particular, Group D had persistently higher rates of ARs than Group C after the first year post LTx (month 12: 24.52% vs. 43.59%, *p* < 0.05; month 18: 24.06% vs. 41.67%, *p* < 0.05; month 24: 21.76% vs. 41.38%, *p* < 0.01). No differences in acute rejection were found between Group B and C, despite a higher prevalence of events in Group B at each evaluation time (Figure 1).

At 24 months post-transplantation, the rate of patients diagnosed with CLAD was different among the groups: group B displayed a higher prevalence of CLAD than the other groups (*p* < 0.05). In particular, a higher number of patients received a diagnosis of CLAD in group B than in group C (50.00% vs. 13.64%; *p* < 0.05) but also when compared with group D (50.00% vs. 0%; *p* < 0.05). Mortality at 24 months after LTx was 9.6%, with no differences among groups.

Considering cases with occurrence of CLAD (groups B and C), patients belonging to group C had a lower incidence of CLAD at 24 months after lung transplantation (*p* = 0.0256) (Figure 2).

Finally, the risk of incidence of at least one AR or CLAD within 24 months of follow-up between patients at intermediate risk (Group B and C) was evaluated (Table 4).

No statistical relevance was found in terms of occurrence of at least one episode of AR between the two groups. The risk of incidence of CLAD after 24 months of follow-up appeared to be higher in Group B patients (*p*: 0.0164; OR: 6.33).

## 4. Discussion

In this retrospective study, we studied different virological data and clinical outcome of 73 LTx patients, including data in relation to QCMV test and CMV serology and evidenced that nonreactivity to QCMV assay in the intermediate-risk population (R+) was associated with an increased risk of acute and chronic rejection at 24 months post-transplantation.

Previous authors demonstrated the protective role of reactive QCMV in terms of occurrence of CMV infections [9]. CMV infection is a risk factor for the occurrence of chronic rejection in lung transplantation [2,16]. In this regard, the QCMV assay has been used in the identification of patients most at risk of CMV viral replication and therefore amenable to more prolonged prophylactic therapy [5,7,8]. Indeed, it has been shown that within populations with moderate-to-severe risk serostatus (R+, D+R−), an unreactive QCMV assay was associated with increased viral replication compared with the test-reactive population [10]. Our study aimed to evaluate whether the QCMV test could be a predictive marker of immunological response to CMV, as well as AR and CLAD, with particular attention to intermediate risk (R+) patients.

According to our findings, CMV replication in lung occurred more frequently in high-risk patients (D+R−, group D) compared to intermediate (R+) and low-risk (D−R−) subjects. Systemic viral replication in whole blood displayed similar occurrence, although with no statistically significant differences. It is interesting to note that low-risk patients (D−/R−) do not show viral activity in the lungs or systemic replication in the blood; this could be responsible for the lower T lymphocytes activation in this population, as reflected by a non-reactive QCMV test. Similar results have been reported in the literature. CMV lung positivity reflects the natural site of viral replication in the alveolar microenvironment [17]. Paraskeva et al. and Cantisan et al. reported the same results with no CMV replication in D−R− LTx individuals [2,18]. On the other hand, an isolated case of viral replication was seen by Johansson in one D−R− LTx and by Bischof et al. in two D−R− kidney transplant patients [19,20]. Nevertheless, in these last studies, it should be noted that CMV-specific prophylaxis was not extended to low-risk patients, differently from our center where combined prophylaxis was demonstrated to be associated with a reduction of AR and lymphocytic bronchiolitis in all LTx [21].

Overall, in our study, no CMV pneumonitis case was detected during follow-up. These results support the literature regarding the protective effect of CMV IgG IV infusion on the onset of CMV viral pneumonia [22,23]. This prophylaxis schedule was administered to all the patients in our study.

The prevalence of ARs among groups followed viral replication in the lungs, as in D+R− patients, CMV manifestations were more frequent. Focusing on intermediate-risk patients, ARs were more frequent among the QCMV non-reactive population, although with no statistical difference. On the contrary, R+Q+ (group C) showed a significantly lower number of events compared to high-risk patients D+R−. A similar comparison between R+Q− patients (Group B) and D+R− patients (group D) displayed no statistical relevance. These data seem to suggest that a reactive QCMV in intermediate-risk patients is protective from the onset of AR, similar to what happens with CLAD incidence.

The connection between CMV replication and the occurrence of AR has been already described in the literature. Roux et al. demonstrated that activation of the immune system following viral infection was linked to AR [3]. Similar data emerged indirectly from a Swedish study that showed a lower incidence of AR in LTx patients receiving prophylactic valganciclovir therapy compared with the same patients receiving oral ganciclovir (characterized by lower drug absorption) due to better control of CMV infection/disease onset [19].

Patients R+ with a non-reactive QCMV assay (Group B) had a higher CLAD incidence. R+ population with a diagnosis of CLAD and a reactive QCMV presented a longer time to CLAD occurrence at 24 months (log rank test, *p*: 0.0256).

Between intermediate risk population, a non-reactive QCMV condition was associated with a higher risk to develop CLAD than reactive QCMV intermediate risk patients (Group C).

A low post-transplant activation of adaptive immunity to CMV leads to an uncontrolled viral replication and, consequently, the elicitation by CMV of shared immune mechanisms in the genesis of CLAD [24,25]. A weak immune response in some patients could be explained by stronger immunosuppression therapy after LTx or by a genetic susceptibility to a weak immune activation. In such non-reactive QCMV patients, some authors suggested reducing the immunosuppressive load or prolonging antiviral prophylaxis [26,27].

This study present some limitations: first, the retrospective nature that could not allow us to draw definitive conclusions. Secondly, the number of enrolled patients was limited, in particular for groups A, B, and C. However, LTx is a rare therapeutic solution worldwide in comparison to other solid organ transplantation; therefore, the number of enrolled patients is of note considering a single transplantation center casuistry. On the other hand, this study, for the first time in the scientific literature of LTx, identifies/describes a new “higher risk” population, i.e., in intermediate risk R+ patients, worthy of particular attention in the post-transplantation follow-up and management. The QCMV test allowed us to detect this population as R+ patients with low adaptive immune system activation against CMV (R+Q−).

Despite the small size of the study population, results were statistically significant and in accordance with the scientific literature regarding both the data of viral replication and the correlation of CMV with acute and chronic rejection.

Lastly, we report no CMV pneumonia cases with a CMV prophylaxis extended to all-risk patients with the use of anti-CMV immunoglobulins prophylaxis, confirming what emerged from previous studies conducted in our center [22].

## 5. Conclusions

Our study reports could add some evidence to the well-known serostatus-based risk classification in lung transplant patients. Nonreactivity to QCMV assay in the intermediate-risk population was associated with an increased risk of chronic rejection at 24 months post-transplantation, and we also identified a new risk category when considering the incidence of acute rejection, which was lower only in the population known to be at higher risk (D+R−). R+ patients with poor activation of adaptive immunity against CMV might therefore be amenable to more serious monitoring and more careful and aggressive prophylactic therapy and/or treatment for CMV viral replication. Potential QCMV monitoring during LTx follow-up should be evaluated.

## Figures and Tables

**Figure 1 viruses-16-01251-f001:**
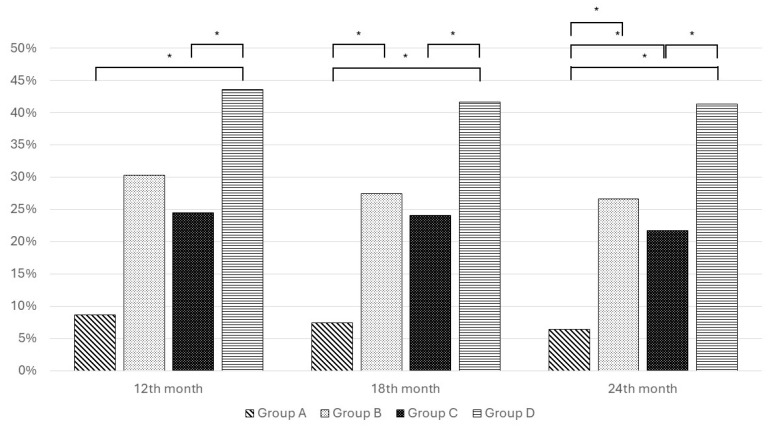
Acute rejection among different groups at 12th, 18th, and 24th month. * *p* < 0.05.

**Figure 2 viruses-16-01251-f002:**
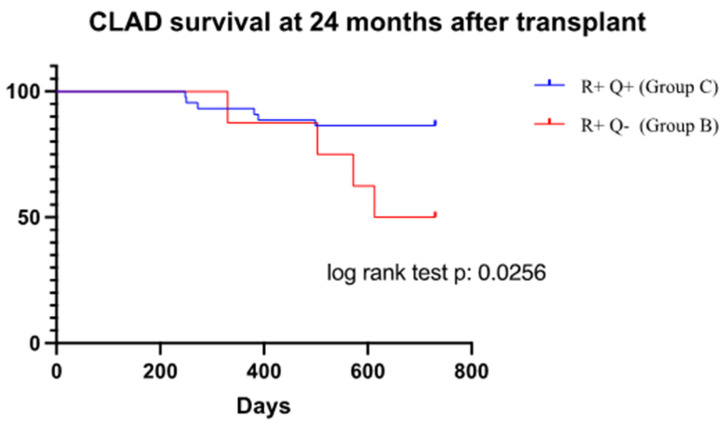
Kaplan–Meier from CLAD at 24 months after transplant (730 days) between groups B (R+Q−) and C (R+Q+).

**Table 1 viruses-16-01251-t001:** Descriptive data of study population divided in four groups. Group A: D−R− with non-reactive QCMV. Group B: R+ with almost two persistent non-reactive QCMV. Group C: R+ with reactive QCMV from the first measurement or following conversion. Group D: D+R−.

	Group A D−R−(Q−)		Group B R+Q−		Group C R+Q+		Group D D+R−(Q−/+)		*p*
Patients, n	6		9		48		10		
M, %	3	50.00%	4	44.44%	35	72.92%	6	60.00%	ns
Age mean, standard deviation	40.50	18.4581	51.44	14.2576	58.08	11.0064	45.4	15.1745	0.0017
QuantiFERON CMV 1st test time (in months, mean)	7.50		5.33		7.22		6		ns
QuantiFERON CMV 1st determination (value in UI/mL, mean)	0.0012		0.036		6.120		1.674		0.030 ^a b d f^
Lung disease preTx									
IPF, %	1	16.67%	1	11.11%	14	29.17%	2	20.00%	ns
NSIP, %	0	0.00%	2	22.22%	5	10.42%	0	0.00%	ns
CF, %	3	50.00%	1	11.11%	7	14.58%	5	50.00%	0.0198
A1ATD, %	0	0.00%	0	0.00%	1	2.08%	0	0.00%	ns
COPD, %	1	16.67%	2	22.22%	12	25.00%	1	10.00%	ns
PH, %	0	0.00%	0	0.00%	5	10.42%	1	10.00%	ns
Others, %	1	16.67%	3	33.33%	4	8.33%	1	10.00%	/
D+, %	/		6	66.67%	34	70.83%	/		ns

Abbreviation list: A1ATD, alpha 1 antitrypsin deficiency; CF, cystic fibrosis; COPD, chronic obstructive pulmonary disease; D, donor; IPF, idiopathic pulmonary fibrosis; M, male; NSIP nonspecific interstitial pneumonia; PH, pulmonary hypertension; Q, CMV QuantiFERON test; ns: no statistical relevance/no analysis. ^a^: *p* < 0.05 A vs. B, ^b^: *p* < 0.05 A vs. C, ^c^: *p* < 0.05 A vs. D, ^d^: *p* < 0.05 B vs. C, ^e^: *p* < 0.05 B vs. D, ^f^: *p* < 0.05 C vs. D.

**Table 2 viruses-16-01251-t002:** Comparison among groups (number of events and their percentages referred to the totality of determinations) about CMV DNA on whole blood and BAL, CMV-therapy, and CMV pneumonitis.

	Group A D−R−(Q−)		Group B R+Q−		Group C R+Q+		Group D D+R−(Q−/+)		*p*
Follow-up until month 12									
CMV DNA blood > 10^5^, %	0/24	0%	0/32	0%	0/182	0%	0/39	0%	/
CMV DNA blood > 10^4^, %	0/24	0%	0/32	0%	3/182	1.65%	2/39	5.13%	ns
CMV DNA blood > 10^3^, %	0/24	0%	3/32	8.82%	17/182	9.34%	8/39	20.51%	ns
CMV DNA BAL > 10^4^, %	0/23	0%	5/32	15.63%	32/168	19.05%	10/38	26.32%	0.0380 ^a b c^
CMV DNA BAL > 10^3^, %	0/23	0%	13/32	40.63%	66/168	39.29%	19/38	50.00%	0.0001 ^a b c^
CMV DNA BAL > 300, %	0/23	0%	15/32	46.88%	80/168	47.62%	23/38	60.53%	0.0028 ^a b c^
CMV-therapy, %	0/24	0%	5/36	13.89%	36/189	19.05%	11/40	27.50%	0.0219 ^b c^
Follow-up until month 18									
CMV DNA blood > 10^5^, %	0/29	0%	0/42	0%	0/223	0%	0/49	0%	/
CMV DNA blood > 10^4^, %	0/29	0%	0/42	0%	3/223	1.35%	2/49	4.08%	ns
CMV DNA blood > 10^3^, %	0/29	0%	4/42	9.52%	18/223	8.07%	8/49	16.33%	ns
CMV DNA BAL > 10^4^, %	0/27	0%	6/41	14.63%	41/204	20.10%	14/48	29.17%	0.0068 ^a b c^
CMV DNA BAL > 10^3^, %	0/27	0%	17/41	41.46%	81/204	39.71%	26/48	54.17%	0.0034 ^a b c^
CMV DNA BAL > 300, %	0/27	0%	20/41	48.78%	97/204	47.55%	31/48	64.58%	<0.0001 ^a b c f^
CMV-therapy, %	0/29	0%	7/45	15.56%	44/233	18.88%	14/50	28.00%	0.0072 ^a b c^
Follow-up until month 24									
CMV DNA blood > 10^5^, %	0/34	0%	0/48	0%	0/257	0%	0	0%	/
CMV DNA blood > 10^4^, %	0/34	0%	0/48	0%	3/257	1.17%	3/59	5.08%	ns
CMV DNA blood > 10^3^, %	0/34	0%	4/48	8.33%	19/257	7.39%	9/59	15.25%	ns
CMV DNA BAL > 10^4^, %	0/31	0%	6/46	13.04%	47/237	19.83%	14/58	24.14%	0.0074 ^a b c^
CMV DNA BAL > 10^3^, %	0/31	0%	18/46	39.13%	94/237	39.66%	29/58	50.00%	0.0014 ^a b c^
CMV DNA BAL > 300, %	0/31	0%	21/46	45.65%	111/237	46.84%	37/58	63.79%	<0.0001 ^a b c f^
CMV-therapy, %	0/34	0%	7/53	13.21%	49/271	18.08%	15/60	25.00%	0.0048 ^a b c^
CMV Pneumonitis, %	0/31	0%	0/46	0%	0/237	0%	0/58	0%	/

Abbreviation list: BAL, bronchoalveolar lavage; CMV, cytomegalovirus; D, donor; R, recipient; Q, CMV QuantiFERON test; ns: no statistical relevance/no analysis. ^a^: *p* < 0.05 A vs. B, ^b^: *p* < 0.05 A vs. C, ^c^: *p* < 0.05 A vs. D, ^d^: *p* < 0.05 B vs. C, ^e^: *p* < 0.05 B vs. D, ^f^: *p* < 0.05 C vs. D.

**Table 3 viruses-16-01251-t003:** Differences among groups in ARs at months 12, 18, and 24 (data are presented as number of events and percentages of events on total determinations). Differences in CLAD and mortality at 24 months.

	Group A D−R−(Q−)		Group B R+Q−		Group C R+Q+		Group D D+R−(Q−/+)		*p*
ARs until month 12, n, %	2/23	8.70%	10/33	30.30%	38/155	24.52%	17/39	43.59%	0.0179 ^c f^
ARs until month 18, n, %	2/27	7.41%	11/40	27.50%	45/187	24.06%	20/48	41.67%	0.0087 ^a c f^
ARs until month 24, n, %	2/31	6.45%	12/45	26.67%	47/216	21.76%	24/58	41.38%	0.0014 ^a b c f^
Patients until month 24,									
n (patients)	5		8		44		10		
CLAD, n, %	0/5	0.00%	4/8	50.00%	6/44	13.64%	0/10	0.00%	0.0377 ^d e^
Death, n, %	0/5	0.00%	0/8	0.00%	7/44	15.91%	0/10	0.00%	ns

Abbreviations list: AR, acute rejection; CLAD, chronic lung allograft dysfunction; D, donor; Q, CMV QuantiFERON test; R, recipient; ns: no statistical relevance/no analysis. ^a^: *p* < 0.05 A vs. B, ^b^: *p* < 0.05 A vs. C, ^c^: *p* < 0.05 A vs. D, ^d^: *p* < 0.05 B vs. C, ^e^: *p* < 0.05 B vs. D, ^f^: *p* < 0.05 C vs. D.

**Table 4 viruses-16-01251-t004:** Comparison between Group B and Group C patients who experienced (or not) at least one AR event and CLAD until 24th month of follow-up.

	**AR**	**noAR**	** *p* **	**OR (IC 95%)**
Group B, n	7	1	0.5147	2.06 (0.23–18.78)
Group C, n	34	10		
	**CLAD**	**noCLAD**	** *p* **	**OR (IC 95%)**
Group B, n	4	4	0.0164	6.33 (1.24–32.38)
Group C, n	6	38		

## Data Availability

Data is unavailable due to ethical restrictions.

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
