# Peer review of "QuantiFERON CMV Test and CMV Serostatus in Lung Transplant: Stratification Risk for Infection, Chronic and Acute Allograft Rejection"

_viruses, 2024, doi:10.3390/v16081251_

Round 1

Reviewer 1 Report

Comments and Suggestions for Authors

The study by Solidoro and colleagues evaluated the use of the QuantiFERON CMV test to predict the risk of CMV manifestations, AR and CLAD in patients after lung transplantation. The authors performed a retrospective study and included 73 patients divided into 4 groups according to CMV serostatus and Quantiferon positivity. The focus was on R+ patients, who are known to be at intermediate risk for uncontrolled CMV replication without prophylaxis/pre-emptive therapy. The authors found that R+ patients with no quantiferon reactivity were associated with an increased risk of developing AR and CLAD. However, I do not see how these conclusions match with the data shown in the tables, clarification would be helpful.

Major points:

-          Table 2:

I could not figure out what the numbers mean? the number of patients or the number of events?

Also, what do the percentages mean? i.e. in the follow-up until month 12th, row 3, column Group B: there are 3 patients with a CMV load >10e3 leading to 8,8%? 3 patients out of 9 patients would be 33%;

This may be a misunderstanding on my part, but clarification would be helpful to understand the statistics.

-          Table 3:

Similar to  Table 2, I could not figure out what the n and % data mean? for AR it seems to me that the „n“ means the events, but what do the percentages mean?

For CLAD, it seems that the „n“ means the number of patients?

Again, this may be a misunderstanding on my part, but clarification would be helpful.

If you use the events for the statistics, then you have to consider how many events belong to how many patients compared to the other groups, otherwise it could be a bias.

-          Quantiferon assay in addition to regular CMV monitoring:

The potential benefit of the quantiferon assay in addition to regular CMV replication monitoring should be evaluated and discussed. Accordingly, it will be important to indicate the optimal time frame after transplantation for testing the reactivity.

In addition, data should be provided showing how often CMV replication was observed prior to testing reactivity with the quantiferon assay. Do the R+ patients who are Q- show any CMV replication?

Minor points:

L29-30: „group B“ and „R+ patients non-reactive QCMV test“ is the same; therefore, the two sentences mean almost the same to me;

L128: two consecutive detections; when were the two consecutive QCMV tests performed? how many days apart? was there a replication episode in between?

L256-257: is this conclusion supported by the data?

 Possible reasons why R+ patients may have a non-reactive QCMV assay should be discussed. In addition, a potential benefit of the QCMV assay in addition to CMV replication monitoring should also be discussed.

Author Response

Thank you for your comments and requests; we provided all the answers and modifications to the manuscript following your indications. 

Major points:

-          Table 2:

I could not figure out what the numbers mean? the number of patients or the number of events? Also, what do the percentages mean? i.e. in the follow-up until month 12th, row 3, column Group B: there are 3 patients with a CMV load >10e3 leading to 8,8%? 3 patients out of 9 patients would be 33%; This may be a misunderstanding on my part, but clarification would be helpful to understand the statistics.

R) Thank you for you comment, we added tot he table text events and percentages, we also added the specifications to patients or events. The numbers and percentages are referred tot he number of events and their percentages on the total of determinations performed in each group.

-          Table 3:

Similar to  Table 2, I could not figure out what the n and % data mean? for AR it seems to me that the „n“ means the events, but what do the percentages mean?

R) Thank you fort he comment. Yes it was the same for table 2. They represents the number of events on the total number of determinations. We specified in the table.

For CLAD, it seems that the „n“ means the number of patients?

R) Yes we specified in the table.

Again, this may be a misunderstanding on my part, but clarification would be helpful.

R) We added the numebr of total patients in each row

If you use the events for the statistics, then you have to consider how many events belong to how many patients compared to the other groups, otherwise it could be a bias.

-          Quantiferon assay in addition to regular CMV monitoring:

The potential benefit of the quantiferon assay in addition to regular CMV replication monitoring should be evaluated and discussed. Accordingly, it will be important to indicate the optimal time frame after transplantation for testing the reactivity.

R) Thank you for these comments but this is a retrospective observational study and we limited our work tot he observations of these events. We performed Quantiferon determination at each follow-up observation and we observed that the response changed during the follow-up (as published by our grou in another work) but we prefer not to suggest any protocol; more specific responses will derive from the prospective ongoing work.

In addition, data should be provided showing how often CMV replication was observed prior to testing reactivity with the quantiferon assay. Do the R+ patients who are Q- show any CMV replication?

R) Sorry but we do not have this data to show.

Minor points:

L29-30: „group B“ and „R+ patients non-reactive QCMV test“ is the same; therefore, the two sentences mean almost the same to me;

R) We fully agree with you. We changed the text oft he abstract.  

L128: two consecutive detections; when were the two consecutive QCMV tests performed? how many days apart? was there a replication episode in between?

R) thank you fort he comment: we added „follow-up“ to the sentence (and so 6 months).

L256-257: is this conclusion supported by the data? R) No this is a speculation, as we wrote „seems to suggest“.  Possible reasons why R+ patients may have a non-reactive QCMV assay should be discussed. In addition, a potential benefit of the QCMV assay in addition to CMV replication monitoring should also be discussed.

R) we did not complete the discussion with the addiction of QCMV assay routine evaluation in follow-up because we focused our attention on discussion of aim-related results; for this reason we added a sentence to the conclusions.

Reviewer 2 Report

Comments and Suggestions for Authors

This study assessed the QuantiFERON CMV test and CMV serostatus in lung transplant. Overall the study is well-designed and the findings had clinical implication. Thus, I just have two minor suggestions.

1. Please correct the way of reference of citation according to the author instruction.

2. Please discuss the limitation of the present study.

3. Please remove the first sentence in the conclusion section.

Author Response

Thank you for your comments and requests; we provided all the answers and modifications to the manuscript following your indications. 

This study assessed the QuantiFERON CMV test and CMV serostatus in lung transplant. Overall the study is well-designed and the findings had clinical implication. Thus, I just have two minor suggestions.

  1. Please correct the way of reference of citation according to the author instruction. R) Thank you, we will modify the text.
  2. Please discuss the limitation of the present study. R) Thank you but limitation of the study are widely discussed.
  3. Please remove the first sentence in the conclusion section. R) Thank you for your comment: we modified the sentence with a less impactful one.

Reviewer 3 Report

Comments and Suggestions for Authors

The manuscript by Solidoro et al. describes a retrospective study on lung transplant recipients, aiming at the question, if the quantification of the CD8+ T-cell response, specific to human cytomegalovirus (HCMV) provides information about the risk for acute and chronic organ rejection.

The reactivation of HCMV poses a major risk for organ rejection and lung disease in the lung transplantation field. The definition of parameters to evaluate the individual risk and to guide further medication is of pivotal importance. The availability of assay formats like the QuantiFERON test enables the quantitative analysis of the T-lymphocyte response against the virus and may thus be an indicator for the risk, posed by the agent.

This study adds to our understanding how the measurement of the CD8+ T-lymphocyte response against HCMV in transplant recipients can be used for risk assessment and to guide clinical proceedings. Given the retrospective nature of the study and the low number of study participants, the findings have to be considered as indicative only. I do have some comments, listed below.

Major

1.       It appears that there were positive QuantiFERON results in the HCMV D-/R- population? The authors should have provided an explanation for this.

2.       The authors should have explained how they standardized the HCMV genome load in BAL specimens.

3.       The data in table 3 regarding acute rejection and chronic lung allograft dysfunction should have been displayed, in addition in a graphical figure to better understand the results.

4.       Lines 201-203: The authors state that CLAD-patients belonging to group C had a better survival rate at 24 months post Tx. However, the table says that 15.91% of patients died in group C but none in group B? This also does not fit to figure 1?

5.       Line 269, The authors claim that the lack of a positive QuantiFERON result “exposes patients to incur in CLAD (by the way faulty expression) almost 6 times more. But table 3 shows a difference between groups B and C of 50% versus 13,64%?

6.       Line 292 ff, the use of antibodies to prevent HCMV reactivation is controversially discussed. As there was no control group on this matter in the manuscript, the role of the antibody prophylaxis cannot be addressed from the data in this study.

Minor

The usage of the English language requires moderate revision (e.g. definite articles are missing at several points, line 88 “preliminary”, “endovenous” etc.)

According to international standards, the authors’ names should be listed using the Christian name first.

Lines 61 and 274, the term “immunization” is misleading, as it should be restricted to the application of vaccines. There is no licensed vaccine directed to HCMV. “Immune response” meant?

Line 131: What is meant by “different results”.

Table 1: What is meant by “1st dosage”?

The text sections related to results should be more clearly separated from the table footnotes.

The lines in figure 1 are barely discernible.

Line 249, citations are at the end of the wrong sentence.

Line 279 f, sentence is incomprehensible

Line 289 f, only some of the data were statistically significant.

Comments on the Quality of English Language

The usage of the English language needs moderate improvement

Author Response

Thank you for your comments and requests; we provided all the answers and modifications to the manuscript following your indications. 

The manuscript by Solidoro et al. describes a retrospective study on lung transplant recipients, aiming at the question, if the quantification of the CD8+ T-cell response, specific to human cytomegalovirus (HCMV) provides information about the risk for acute and chronic organ rejection.

The reactivation of HCMV poses a major risk for organ rejection and lung disease in the lung transplantation field. The definition of parameters to evaluate the individual risk and to guide further medication is of pivotal importance. The availability of assay formats like the QuantiFERON test enables the quantitative analysis of the T-lymphocyte response against the virus and may thus be an indicator for the risk, posed by the agent.

This study adds to our understanding how the measurement of the CD8+ T-lymphocyte response against HCMV in transplant recipients can be used for risk assessment and to guide clinical proceedings. Given the retrospective nature of the study and the low number of study participants, the findings have to be considered as indicative only. I do have some comments, listed below.

Major

  1. It appears that there were positive QuantiFERON results in the HCMV D-/R- population? The authors should have provided an explanation for this. R) This is not correct because our D-/R- population (Group A) has no D+ patients (as explained in Table 1, last row). The Quantiferon mean dosage 0.0012 is the explanation.
  2. The authors should have explained how they standardized the HCMV genome load in BAL specimens. R) Thank you, we reported the reference where it was detailed.
  3. The data in table 3 regarding acute rejection and chronic lung allograft dysfunction should have been displayed, in addition in a graphical figure to better understand the results. R) we modified the numbers presented in table 3 to make the table easier to understand; moreover, we added a new figure.
  4. Lines 201-203: The authors state that CLAD-patients belonging to group C had a better survival rate at 24 months post Tx. However, the table says that 15.91% of patients died in group C but none in group B? This also does not fit to figure 1? R) Thank you, your comment is correct. We make an error, and we corrected the text as well as the figure legend. We also corrected the discussion.
  5. Line 269, The authors claim that the lack of a positive QuantiFERON result “exposes patients to incur in CLAD (by the way faulty expression) almost 6 times more. But table 3 shows a difference between groups B and C of 50% versus 13,64%? R) The six time more was derived from the OR calculated and presented in Table 4. To obviate to this possible misunderstanding we decided to modify the sentence, talking about a general increased risk to develop CLAD.
  6. Line 292 ff, the use of antibodies to prevent HCMV reactivation is controversially discussed. As there was no control group on this matter in the manuscript, the role of the antibody prophylaxis cannot be addressed from the data in this study. R) We fully agree with you. We modified the text.

Minor

The usage of the English language requires moderate revision (e.g. definite articles are missing at several points, line 88 “preliminary”, “endovenous” etc.). R) We provided an extended English revision of the entire manuscript.

According to international standards, the authors’ names should be listed using the Christian name first. R) We modified the list

Lines 61 and 274, the term “immunization” is misleading, as it should be restricted to the application of vaccines. There is no licensed vaccine directed to HCMV. “Immune response” meant? R) Yes we modified the text.

Line 131: What is meant by “different results”. R) QCMV indifferently non-reactive or reactive; we changed the text.

Table 1: What is meant by “1st dosage”? R) Determination, we changed the text

The text sections related to results should be more clearly separated from the table footnotes. R) This is an editing problem that will be solved during the final editing phase. 

The lines in figure 1 are barely discernible. R) This is a problem that will be solved during the final editing of the paper.

Line 249, citations are at the end of the wrong sentence. R) we moved citations

Line 279 f, sentence is incomprehensible. R) We meant that in these cases some authors suggest reducing the immunosuppressive load (by reducing the dosage of immunosuppressants). We modified the text.

Line 289 f, only some of the data were statistically significant. R) We agree with you but the results have been discussed above in the text and this is a very brief resume in the “limits of the study” section.
